# Development of a High-Sensitivity Millimeter-Wave Radar Imaging System for Non-Destructive Testing

**DOI:** 10.3390/s24154781

**Published:** 2024-07-23

**Authors:** Hironaru Murakami, Taiga Fukuda, Hiroshi Otera, Hiroyuki Kamo, Akito Miyoshi

**Affiliations:** 1Institute of Laser Engineering, Osaka University, 2-6 Yamadaoka, Suita 565-0871, Osaka, Japan; 2TAIYO YUDEN CO., LTD., 8-1 Sakaemachi, Takasaki 370-8522, Gunma, Japan

**Keywords:** millimeter wave, SAR, MIMO, imaging, NDT, radar module

## Abstract

There is an urgent need to develop non-destructive testing (NDT) methods for infrastructure facilities and residences, etc., where human lives are at stake, to prevent collapse due to aging or natural disasters such as earthquakes before they occur. In such inspections, it is desirable to develop a remote, non-contact, non-destructive inspection method that can inspect cracks as small as 0.1 mm on the surface of a structure and damage inside and on the surface of the structure that cannot be seen by the human eye with high sensitivity, while ensuring the safety of the engineers inspecting the structure. Based on this perspective, we developed a radar module (absolute gain of the transmitting antenna: 13.5 dB; absolute gain of the receiving antenna: 14.5 dB) with very high directivity and minimal loss in the signal transmission path between the radar chip and the array antenna, using our previously developed technology. A single-input, multiple-output (SIMO) synthetic aperture radar (SAR) imaging system was developed using this module. As a result of various performance evaluations using this system, we were able to demonstrate that this system has a performance that fully satisfies the abovementioned indices. First, the SNR in millimeter-wave (MM-wave) imaging was improved by 5.4 dB compared to the previously constructed imaging system using the IWR1443BOOST EVM, even though the measured distance was 2.66 times longer. As a specific example of the results of measurements on infrastructure facilities, the system successfully observed cracks as small as 0.1 mm in concrete materials hidden under glass fiber-reinforced tape and black acrylic paint. In this case, measurements were also made from a distance of about 3 m to meet the remote observation requirements, but the radar module with its high-directivity and high-gain antenna proved to be more sensitive in detecting crack structures than measurements made from a distance of 780 mm. In order to estimate the penetration length of MM waves into concrete, an experiment was conducted to measure the penetration of MM waves through a thin concrete slab with a thickness of 3.7 mm. As a result, Λ_exp_ = 6.0 mm was obtained as the attenuation distance of MM waves in the concrete slab used. In addition, transmission measurement experiments using a composite material consisting of ceramic tiles and fireproof board, which is a component of a house, and experiments using composite plywood, which is used as a general housing construction material in Japan, succeeded in making perspective observations of defects in the internal structure, etc., which are invisible to the human eye.

## 1. Introduction

In recent years, the maintenance and inspection of aging social infrastructure in public facilities, dams, tunnels, etc., has become a critical issue due to the aging of structures and the increasing need for safety and reliability. Traditionally effective inspection methods often involve invasive procedures that can be time-consuming, costly, and potentially damaging to the structures being inspected. As a result, there is a growing demand for non-destructive testing (NDT) methods that can provide accurate and efficient assessments without compromising the integrity of the infrastructure.

The advancement of NDT technologies for infrastructure and residential buildings is particularly important in Japan, a country highly prone to earthquakes. The ability to accurately assess and ensure the integrity of structures without causing additional damage is critical to mitigating the risks associated with seismic activity and maintaining public safety.

Several NDT techniques are available for such aging infrastructure facilities. Typical examples include ultrasonic testing (UT) [1,2], thermographic testing (TT) [3,4,5], X-ray testing (XT) [6], pulsed magnetic testing (PMT) [7,8], etc. UT can detect internal defects with high accuracy and is often portable but requires contact with the test object and ultrasound couplants. Thermographic testing is non-contact and can inspect large areas in a short time, but is affected by changes in surface temperature, so internal defects can be difficult to detect. XT can image internal structures using transmitted X-rays and can produce high-resolution images, but requires control of radiation, which is harmful to the human body and is often a more bulky instrument. MPT can also detect microscopic defects near the surface, but the measurement is limited to magnetic materials and requires cleaning of the surface of the object being inspected.

A promising technology in the field of NDT is millimeter-wave (MM-wave) radar imaging. Operating in the frequency range of 30 to 300 GHz, MM-wave radar provides high-resolution imaging and the ability to penetrate various materials, making it particularly suitable for the detection of cracks on the surface of steel bridges and concrete tunnels, and for inspecting houses damaged by earthquakes and other disasters. This technology offers several advantages over traditional methods, including the ability to perform inspections in real-time, detect subsurface defects, and operate in adverse environmental conditions. In addition, if the directivity of MM-wave radar can be increased sufficiently, it is possible to inspect measurement objects located at a considerable distance. This is an important factor in ensuring the safety of the technician conducting the inspection.

However, until now, research on non-destructive testing using MM-wave has shown the following:(a)Highly sensitive but large systems using a vector network analyzer, waveguide cables, horn antennas, etc.: Expensive and requires skilled technology [9,10].(b)Inspection using a handy type of imaging system: Observation in contact conditions is essential because of the use of near fields [11,12].(c)Inspection using a compact MIMO (multiple-input and multiple-output)-SAR (synthetic aperture radar) imaging system with an MM-wave radar module of one board type: Remote inspection is possible, but commercially available cost-effective radar modules lack sensitivity [13,14].

Therefore, in this study, a highly sensitive MM-wave radar array antenna module with high directivity and high signal-to-noise ratio (SNR) was developed and integrated into a SIMO (single-input and multiple-output)-SAR imaging system. After optimizing the imaging performance of the constructed system, the following experiments were performed:(1)The SNR was compared with the previously built MIMO SAR system using the Texas Instruments IWR1443BOOST Evaluation Module (EVM) [15], and an increase in sensitivity of about 5.4 dB was confirmed.(2)The measurement of ~0.1 mm wide cracks in a concrete block covered with glass fiber-reinforced (GFR) tape and acrylic black paint to prevent visibility clearly showed the small 0.1 mm crack.(3)Reinforced concrete slabs with embedded steel frames were made and observed. The results showed a small amount of the steel frame near the surface layer, and from subsequent transmission measurements using a thin concrete slab of 3.7 mm thickness, a penetration length of 6 mm could be estimated for MM waves.(4)Transmission image measurements of ceramic tiles, a refractory board, and plywood, which are used as building materials for ordinary dwellings, were performed to confirm the defects inside these materials.

These results demonstrate the usefulness of the system developed in this study for non-destructive and non-contact inspection. These details are reported.

## 2. MM-Wave Array Antenna Sensor

To explore the possibility of MM-wave imaging for NDT, the transmission characteristics of MM-wave against various materials that would attenuate MM waves should be investigated. For this purpose, an MM-wave array antenna module with high absolute gains of the transmit and receive antennas was prepared for the present studies. Specifically, a two-dimensional orthogonal array antenna sensor was fabricated by TAIYO YUDEN CO., LTD. (Japan). The array pattern of the transmit and receive antennas is shown in Figure 1, which consists of twelve transmit antennas (Tx) with a spacing of λ (=3.80 mm) and sixteen receive antennas (Rx) with a spacing of 0.86λ (=3.25 mm) in an antenna unit of 52.5 mm × 73.6 mm × 19.4 mm, in an orthogonal array pattern of MIMO antenna structure. Although these antennas are arranged in such a narrow pitch, they have excellent side-rope suppression and can control directivity over a wide bandwidth. In addition, there is very low signal transmission loss between these Tx and Rx antenna elements and the radar chips using the technology developed by Aoki et al. [16]. For the MM-wave radar chips, four cascaded chips (AWR2243 marketed by Texas Instruments [17]), which are FMCW transceivers in the 76.5 GHz to 81 GHz band, are installed on this antenna unit. The element statistics of the MM-wave array antenna module are shown in Table 1.

Looking at the directivity and absolute gains, which are the most important factors in the present study, it can be seen that despite such a small antenna device, the directivity is high enough to achieve an absolute gain of 13.5 dB and 14.5 dB for the Tx and Rx antenna elements, respectively. These values are significantly higher than the 10.5 dB of the Texas Instruments IWR1443BOOST Evaluation Module (EVM) [15] used in our previous MM-wave MIMO-SAR imaging system [13]. In addition, the elevation 3 dB beamwidth is approximately ±13 degrees, indicating the excellent directivity of the present radar module.

## 3. SIMO-SAR Imaging System

To perform high-resolution imaging, a SAR imaging system was constructed. SAR uses the concept of a virtual orbit, where a large number of antennas are distributed along the path of the radar system, allowing for signal transmission and reception while the system is moving [18,19,20,21,22]. This dynamic motion effectively increases the aperture length, thereby improving system performance. In this system, only a Tx7 antenna was used, as shown in Figure 1. In other words, a SIMO-SAR imaging system was constructed. On the other hand, a two-axis mechanical scanner (YAMAHA HXYx series) was adopted for SAR imaging. The maximum scanning range (synthetic aperture) in the horizontal direction ($L_{SA\_X}$) is 1250 mm with a maximum speed of 1200 mm/s, while that in the vertical direction ($L_{SA\_Y}$) is 1050 mm with a maximum speed of 600 mm/s. Control was performed in a Python-based environment from a host PC via a serial port.

Since there is very little interference between the sixteen Rx antennas, even though the Rx spacing is very small, 16 rows of data in the *Y*-axis direction can be acquired in a single scan along the *X*-axis. However, for high-resolution imaging in the *Y*-axis direction, the *X*-axis scanner was shifted by 0.5∆ (∆ = 0.86 *λ*: 3.25 mm) in the *Y*-axis direction and scanned again along the *X*-axis before the *X*-axis scanner was shifted by 15.5∆ in the *Y*-axis direction (see Figure 2a).

For the FMCW chirp signal, the frequency of the signal transmitted from the Tx antenna was varied from 76.5 GHz to 81 GHz (bandwidth *B* = 4 GHz) based on the effective value, and the frequency slope in the chirp signal was set to 28.0 MHz/μs. The horizontal (*X*-axis) scanning speed was set to 500 mm/s, the sampling interval in the scanning along the *X*-axis was set to 2.0 ms, and the effective number of samples was set to 768.

In FMCW radar measurement, the received signal reflected from a target is mixed with the transmitted signal by a mixer, and the differential frequency components of the two signals are output through a low-pass filter [23]. This extracts an intermediate frequency (IF) signal [24,25]. The IF signal can be used to obtain a range FFT spectrum in which each spectrum peak reflects the distance (*L*) of the target surfaces. Therefore, it is possible to obtain 2D and 3D reconstructed images using the range FFT spectrum. The detailed data process for obtaining the 2D or 3D reconstructed images using the IF signal has already been reported elsewhere [13].

## 4. Basic Performance of the Constructed SIMO-SAR Imaging System

### 4.1. Optimization of Millimeter-Wave Imaging Measurements

In the above, we described the details of the SIMO-SAR imaging system constructed this time. However, since the measurements were performed in a laboratory environment with limited space, the measurement conditions for the present study had to be modified somewhat. In particular, an MM-wave array antenna module with such large absolute gains, a high directivity, and up to sixteen Rx antennas probably has no problems measuring distant targets. However, there are some considerations when measuring in a limited laboratory space. In particular, the distances between T7 and R1 (6 mm) and between T7 and R16 (58 mm) are too different, which means that if the distance to the target to be measured is short, there will be a large difference in the intensity of the MM-wave signals received by the Rx1 and Rx16 antennas.

To investigate this point, we compared the MM-wave images of a copper plate placed at different distances. The results are shown in Figure 3. Figure 3a shows a 200 mm × 200 mm copper plate used for this MM-wave imaging. This copper plate was placed in front of the constructed SIMO-SAR imaging system at different distances of *L* = 720 mm and *L* = 1850 mm in the measurements. The left sides of Figure 3b,c shows the MM-wave images observed when the copper plate was placed at *L* = 720 mm and *L* = 1850 mm, respectively. The right sides of Figure 3a,b shows the *Y*-axis profile of the averaged intensity $\bar{I}$ values within the red dotted rectangles inserted in the MM-wave images. The intensity (*I*) corresponds to the MM-wave power received by the receive antenna. All receive antennas of Rx1~Rx16 were used in the measurements. Here, the difference in the MM-wave images between Figure 3b,c clearly shows that the difference in the distance *L* from the transmit antenna Tx7 causes a significant difference in the signal intensities received by the Rx1~Rx16 antennas when observed at a short distance of *L* = 720 mm. In particular, the profile in Figure 3b shows apparent areas where the reflection signal from the target can hardly be received.

The result obtained at *L* = 720 mm can be well explained by considering the situation as shown in Figure 4. When the distance *L* = 720 mm, the elevation angle θ[L720,d57] from Tx7 of the point on the target located in front of Rx16 (distance *d* between Tx7 and Rx16: 58 mm), which is the farthest from Tx7, is 4.6 degrees. Considering the beamwidth given in Table 1, the MM-wave power at the point on the target in front of Rx16 would be about 1 dB less than the absolute gain of 13.5 dB. In addition, MM waves are reflected from flat metal surfaces of copper plate with a reflection angle nearly equal to the angle of incidence. Therefore, it is expected that MM waves will hardly be received by the Rx16 antenna. Increasing the distance *L* improves this situation somewhat, but there should be still a significant difference in the received signal intensity between Rx1 and Rx16, as shown in Figure 3c. Based on these results, we decided to use only Rx1 to Rx8 antennas as the receive antennas in the present study, according to the diagram of the *X*-*Y* scanning diagram, as shown in Figure 2b. Figure 3d shows the result of the measurement with the number of antennas reduced while the distance *L* remains at 1820 mm. It can be seen that the change in the signal intensity is considerably improved. In addition, in the present study, the measurements were performed with distant targets as far as possible.

### 4.2. Signal-to-Noise Ratio of the Constructed System

Furthermore, the SNR value of the present SIMO-SAR imaging system (system-A) was compared with that of the previously constructed MIMO-SAR imaging system with an IWR1443BOOST EVM (system-B) [13]. The IWR1443BOOST EVM has an absolute gain of approximately 10.85 dB over the 76 to 81 GHz frequency band and an elevation 3 dB beamwidth of approximately ±28 degrees [15]. For comparison, MM-wave images of the copper plate were observed by both systems. Each distance *L* from the radar module to the copper plate was *L*_A_ = 1810 mm for system-A and *L*_B_ = 680 mm for system-B. The results are shown in Figure 5a,b, where the top figure shows the MM-wave image observed for each system, while the bottom figure shows the profile of the averaged signal intensity $\bar{I}$ along the *X*-axis direction in the area indicated by the dashed red rectangle inserted in the top image. From these results, the averaged signal intensity ($\bar{I_{S}}$) and noise intensity ($\bar{I_{N}}$) were estimated as follows:(1)$${SNR}_{A}=10{log}_{10}\left(\frac{\bar{I_{S}}}{\bar{I_{N}}}\right).$$

As a result, we obtained the following SNR values for system-A (SNR_A_) and system-B (SNR_B_):(2)$${SNR}_{A}\;≅\;10{log}_{10}\left(\frac{836}{16.6}\right)=17.0\;dB.$$
(3)$${SNR}_{B}≅10{log}_{10}\left(\frac{355}{24.4}\right)=11.6\;dB.$$

Note that the observed SNR_A_ is 5.4 dB higher than the SNR_B_. Considering the radar equation [26], it is expected that the received power using system-A will be significantly attenuated because the distance *L*_A_ is about 2.66 times longer than *L*_B_ for system-B. Despite this fact, the fact that system-A has a higher SNR value than system-B is discussed below.

When considering the difference in SNR between the two systems, we should first consider the gain as a function of the number of array elements *n*. In the measurements, system-A used eight array elements (*n* = 8: Tx-1ch × Rx-8ch), and system-B with the IWR1443 BOOST EVM also used eight array elements (*n* = 8: Tx-2ch × Rx-4ch). Therefore, the gain (*G*_array_) for both radar modules is given as follows:(4)$$G_{array}=10\log\left(n\right)=10\log\left(9\right)=9.5\;dB.$$

Since there is no difference in the gain $G_{array}$, the difference in the composite gain for the two systems was considered using the radar received power equation [26].

Assume that the results of Figure 5 were obtained under the same condition of radar scattering cross section *σ* and the wavelength *λ* and that the difference in SNR values is considered to be given by the ratio of the maximum received power of each radar module of system-A and system-B. Therefore, the difference in the SNR value of 5.4 dB would be expressed by using the receive powers of *P*_RA_ and *P*_RB_ and the emission peak powers of *P*_TA_ and *P*_TB_ for system-A and system-B, respectively, as follows:(5)$$5.4\;dB=10log\frac{P_{RA}}{P_{RB}}=\left(G_{TxA}+G_{RxA}\right)-\left(G_{TxB}+G_{RxB}\right)+10log\left[\frac{\left\{\frac{\sigma \lambda^{2}P_{TA}}{{\left(4\pi \right)}^{3}{\left(L_{A}\right)}^{4}}\right\}}{\left\{\frac{\sigma \lambda^{2}P_{TB}}{{\left(4\pi \right)}^{3}{\left(L_{B}\right)}^{4}}\right\}}\right]$$

Substituting the antenna gains for each module yields the following equation.
(6)$$10log\frac{P_{RA}}{P_{RB}}=28\;dB-21.7\;dB+10log\left[{\left(\frac{L_{B}}{L_{A}}\right)}^{4}\times \frac{P_{TA}}{P_{TB}}\right].$$

Here, 10.85 dB was used as $G_{TxB}$ and $G_{RxB}$ for the IWR1443BOOST EVM [15]. Therefore, the ratio of emission peak powers of each radar module can be obtained as follows:(7)$$10log\left(\frac{P_{TA}}{P_{TB}}\right)=5.4\;dB-6.3\;dB-10log{\left(\frac{L_{B}}{L_{A}}\right)}^{4}=-0.9\;dB-40log\left(\frac{680}{1810}\right)=16.1\;dB$$
(8)$$\frac{P_{TA}}{P_{TB}}={\;10}^{1.61}\;(=40.7)$$

The present radar module with four cascaded chips (Texas Instruments AWR2243) is indeed considered to have a high transmission power, but it is not possible to have 40 times more transmission power than that of the IWR1443 BOOST EVM. Therefore, this result strongly indicates a very low signal transmission loss between the Tx and Rx array antenna elements and the four cascaded chips and also indicates a very high directivity of the present radar module.

## 5. Experimental Results on Transmission Performance of MM Wave and Discussion

### 5.1. Comparative Studies on Crack Detection in Concrete

Based on the results of the preliminary experiments, the potential of the present imaging system for NDT inspection was investigated. First, a comparison study with system- B was conducted to detect the cracks between concrete blocks under glass fiber-reinforced (GFR) tape and a coat of black acrylic paint. The results of this experiment are shown in Figure 6a through Figure 6d. As shown in Figure 6a, two concrete blocks were firmly bonded together, and GFR tape and a coat of black acrylic paint were applied over a portion of the cracks to hide the crack areas. Figure 6b is a magnified photograph taken to show the width of the cracks. As can be seen from this photograph, the width of the crack is approximately 0.1 mm. Figure 6c shows the MM-wave image observed with system-A. It can be seen that the cracks are visible in the image below the GFR-taped areas. On the other hand, Figure 6d shows the result with system-B. In this case, the coating paint itself is transparent, but the GFR tape is not sufficiently transparent and the cracks in the underlying concrete are not visible.

The detection of cracks caused on the surface of concrete infrastructure facilities is one of the most important issues in NDT, and as shown in Figure 6c,d, we can observe the cracks on the exposed surface areas of concrete blocks. On the surface of the concrete blocks, which does not have a smooth surface like a copper plate, the incident MM waves are scattered uniformly in all directions and reflected with some intensity (backscattering) in the direction of incidence. As a result, unlike objects with smooth surfaces, an overall dark image is observed. However, in the areas with cracks, since the incident MM waves are scattered more strongly and irregularly, the intensity of refracted MM waves returning in the directions of the receive antennas would be much smaller, which would be reflected as darker lines in the MM-wave image.

On the other hand, in the area with GFR tape and black acrylic paint on the surface, we could see cracks hidden under these coverings only with the present system-A with high sensitivity. Incidentally, the contrast in the raw photographic data was adjusted so that the viewer could see where the GFR tape was applied in Figure 6a. From these two MM-wave images, there is almost no change in the received signal intensity for the black acrylic paint, but the signal intensities are weaker in the areas of the GFR tape, confirming that absorption by the GFR tape has occurred. To be sure, contrast-adjusted photographs are shown in the upper left corner of Figure 6c,d. Here, the contrast has been adjusted to show the crack structure most clearly. As can be seen from this photograph in Figure 6c, the MM wave image observed using system-B did not show the crack structure hidden under the GFR tape, no matter what adjustments were made.

We also tried to make measurements under more severe conditions that would reduce the spatial resolution and signal intensity. In this experiment, measurements were made with a fairly thick top coat of acrylic paint and at a distance of *L*_A_ = 3038 mm from the radar to the concrete blocks. For comparison, a crack-free concrete block with GFR tape attached to the bottom half of the block and a thick layer of acrylic paint applied over it was also prepared for simultaneous measurement. Figure 6e shows a photograph of these two targets placed at *L*_A_ = 3038 mm. Figure 6f shows the MM-wave reflection images observed at these target surface locations. In the MM-wave image observed on the cracked concrete blocks, the spatial resolution appears to be slightly lower, but the cracks and chips in the block are more clearly visible than in the case of *L*_A_ = 780 mm. Furthermore, it can be seen that no such traces were observed in the concrete blocks without cracks. Although the concrete surfaces of infrastructure facilities are generally protected by reinforcing sheets and paint in many cases, the highly sensitive MM-wave imaging system developed in this study proved to be sufficiently useful for detecting cracks that are not visible on such objects.

Let us now consider why 0.1 mm wide cracks can be detected using MM waves with a wavelength of about 4 mm. According to Ref. [27], the horizontal and vertical cross-range resolutions $\delta_{X}$ and $\delta_{Y}$ in SAR imaging are given as follows:(9)$$\delta_{X}\;≅\;\frac{\lambda_{c}L_{A}}{2L_{{SA}_{X}}}=\frac{3.80\times L_{A}}{2\times 1250}\left[mm\right],$$
(10)$$\delta_{Y}\;≅\;\frac{\lambda_{c}L_{A}}{2L_{{SA}_{Y}}}=\frac{3.80\times L_{A}}{2\times 1050}\left[mm\right],$$
where $\lambda_{c}$ = 3.80 mm is the wavelength at the center frequency of 78.8 GHz in the chirp signal. Therefore, considering the case of *L*_A_ = 780 mm, the following values can be obtained as the cross-range resolution: $\delta_{X}$ = 1.19 mm and $\delta_{Y}$ = 1.41 mm. As for $\delta_{Y}$, it cannot be less than 0.5∆ (=1.4 mm), which is the minimum vertical motion interval of the receive antenna Rx. Similarly, for *L*_A_ = 3038 mm, $\delta_{X}$ is 4.62 mm. Looking at the $\delta_{X}$ values, both are very large compared to the actual crack width of 0.1 mm, which would make it difficult to detect such narrow cracks, but they are observed. In general, if there are cracks or defects on the surface of an object, the surface irregularities will interfere with the reflection of the MM waves, changing the scattering pattern and reducing the radar cross-section. This causes the received signal strength to decrease in the range of $\lambda_{c}$ around the crack or defect, and if this change can be detected with high sensitivity, the presence or absence and location of the defect can be confirmed. Therefore, although a spatial resolution of about one wavelength is necessary for non-destructive and non-contact inspection using MM waves, SNR is considered more important. One of the advantages of MM-wave imaging is that microdefects and cracks as small as 0.1 mm can be observed as an image spread over a few millimeters of the wavelength of the MM wave. This is because non-destructive inspections of infrastructure facilities are generally performed over a relatively large area of several meters, and if, for example, point or line defects with a spread of several millimeters (>1/1000) appear in an inspection image of an area of several meters squared, the possibility of the presence of a defect can be immediately suspected. On the other hand, in inspections using visible light, which has a short wavelength and allows for high-resolution observation, defects that are larger than the wavelength of the light (<1 µm) will appear in the observed image as their actual size. In other words, defects as small as about 0.1 mm (<1/10,000) will appear as dots or lines on an inspection image of an area of several square meters. It is considered extremely difficult to detect the presence or absence of minute defects in such an observed image.

The validity of such an idea is evident from the crack images in the MM-wave images in Figure 6c,f. Comparing the crack images in Figure 6c,f, it is obvious that the image in Figure 6c shows a finer crack structure. However, in terms of signal intensity and contrast of these images, the signal intensity is higher, and the presence of cracks can be observed more clearly in Figure 6f, even though the measurement distance is about four times longer. At a distance of *L* = 3038 mm, the elevation angle from the transmit antenna Tx7 to the point in front of the receive antenna Rx8 on the concrete surface is only 0.6 degrees. Thus, it can be seen that both transmission and reception are carried out using a signal of almost maximum gain. This indicates that the newly developed radar module with high directivity and high gain is very useful for non-destructive, non-contact, and safer remote inspection of concrete structures at risk of collapse due to aging or natural disasters. Furthermore, it can be seen that in this type of non-destructive testing using MM waves, it is necessary to develop a compact system with signal strength rather than resolution, i.e., high SNR.

Another interesting result is that although the observation with system-A was made by placing the concrete blocks at a very close distance of *L*_A_ = 780 mm from the radar module, no significant difference in signal intensities appeared, as shown in the MM-wave images of a copper plate. Although a periodic dark and light wide stripe structure is observed approximately equal to the total length of the array of eight receiving antennas (8∆ = 26 mm), the signal is not extremely weak, as shown in Figure 3b,c. This is because the surface of the concrete blocks is not perfectly smooth, and the incident waves scattered by the surface are reflected to some extent in the direction of the receive antenna and detected. This indicates that the MM-wave radar inspection method is fully applicable to the non-destructive inspection of structures with such surfaces.

In this section, we investigated the detection of surface cracks that may occur in common concrete structures such as tunnels and buildings. The next section reports the results of experiments conducted on a concrete slab and a thin concrete slab to evaluate the transmission performance of MM waves against concrete.

### 5.2. Evaluation of Transmission Performance Using a Concrete Slab

To evaluate the performance of MM-wave transmission to concrete, a reinforced concrete (RC) slab with a steel frame of D13 (ϕ12.7 mm) was prepared, as shown in Figure 7. It is difficult to see here, but steel frames #1 and #5~#8 are straight, and the other steel frames #2~#4 are bent near the center. The amount of bending increases from #2 to #4. The assembled steel frames were placed in wooden frames and poured with concrete, and this RC slab was removed and measured after the concrete had sufficiently dried and hardened.

Figure 8 shows the photograph of the RC slab and its MM-wave images. Figure 8a shows the surface of the measured RC slab. The measurements were made on the surface corresponding to the bottom of the RC slab in Figure 7. It can be seen that the entire steel frame of #1 is visible, while the central regions of steel frames #2~#4 are partially buried under the concrete. Figure 8b–d shows the MM-wave images observed at a distance of *L* = 1000 mm, *L* = 2000 mm, and *L* = 3050 mm, respectively. The horizontal stripe structure of light and dark observed at *L* = 1000 mm disappears as the distance *L* increases from *L* = 1000 mm to *L* = 3050 mm. In addition, the steel frame of #3, which is moderately bent among the steel frames of #2~#4, becomes more visible in the MM-wave images with increasing distance. However, only the two ends are weakly imaged for the steel frame of #4, which has the largest bending even at the distance of *L* = 3050 mm. This is because the incident waves are attenuated inside the concrete, and the reflected wave from the inclined targets hardly returns to the receive antenna. In addition, we could not observe anything at all about steel frames #5~#8, which are mounted under steel frames #1~#4 and buried at least 13 mm under the concrete surface. This is discussed in the next section by estimating the attenuation distance Λ of MM waves in concrete.

On the other hand, we can see more detailed structures of the steel frame of #1 in the MM-wave image, as shown in the inset photo of Figure 8d. The MM-wave image in Figure 8d shows the most sensitive and clear image, reflecting the steel frames inside the concrete. Calculating the elevation angle θ[*L*3050, *d*32] from Tx7 to the target in front of Rx8, it is about 0.6 degrees. This result indicates that the measurement was made using the central regions with the highest beam power. In general, non-destructive and non-contact inspection is highly desirable in any kind of inspection of infrastructure facilities and residential buildings, because such inspection can ensure the life safety of measurement technicians, especially when inspecting buildings after a major earthquake.

### 5.3. Estimation of Attenuation Distance of MM Waves to Concrete26

To discuss the observed results on the RC slab, let us estimate the attenuation distance Λ of MM waves to concrete materials by using basic electromagnetic formulas [28]. The real part of the complex permittivity $\varepsilon =\varepsilon_{1}+i\varepsilon_{2}$ in the MM-wave frequency range is reported to be $\varepsilon_{1}$ = 5~10 and $\varepsilon_{2}$ = 0.2~1 [29,30,31,32]. Therefore, if we calculate the complex refractive index N=n+iκ using the intermediate values $\varepsilon_{1}=7.5$ and $\varepsilon_{2}=0.6$, we can estimate n=2.74 and κ=0.11. Using these values of n and κ, we can estimate the reflectance *R* of MM waves under the condition of the air–concrete interface in the normal incidence of MM waves. First, the reflectance *R* is estimated to be about 22%. In addition, the absorption coefficient α (and attenuation distance Λ) in concrete is estimated to be α=0.18 mm^−1^ (and Λ=5.4 mm) at 80 GHz. Here, the attenuation distance Λ=5.4 mm explains well why the steel frames #5~#8 could not appear in the MM-wave images at all.

Based on the estimated Λ value of 5.4 mm, we prepared a concrete slab of about 3.7 mm thickness and attempted transmission measurements to demonstrate the actual transmission performance of MM waves to a concrete slab.

### 5.4. Evaluation of Transmission Performance Using a Thin Concrete Slab

While all MM-wave images presented so far have been single reflection images of the target, the SAR imaging technique using FMCW radar converts the obtained IF signal into a range FFT signal, so that a 2D cross-sectional image (2D slice image) can be obtained at any distance in front of the MM-wave radar module [13]. Therefore, if there is some target of interest behind an object through which MM waves can penetrate, the reflection image of the target can also be observed. Therefore, as shown in Figure 9a, a thin concrete slab with a thickness of 3.7 mm was prepared and placed at *L* = 3100 mm in front of the radar module, and a copper plate was placed at *L* = 3185 mm behind it, and the transmission performance of MM waves through concrete was investigated by observing the reflection image of the copper plate. Figure 9b shows the MM-wave image of the concrete surface at *L* = 3100 mm, and Figure 9c shows the MM-wave image at *L* = 3185 mm corresponding to the surface of the copper plate behind the concrete slab. As can be seen from Figure 9a, the uppermost part of the copper plate protrudes from the concrete slab and is exposed, so Figure 9c shows that the signal intensity in this exposed area is considerably higher than in other areas. Therefore, the attenuation distance of the concrete was calculated by comparing the received signal intensity from the uppermost part of the copper plate and the hidden part behind the concrete in this experiment.

To compare the signal intensities from both parts, the averaged signal intensities along the *X*-axis direction in area-1 and area-2 in Figure 10a were used. The distribution of the averaged intensity $\bar{I}$ in area-1 and area-2 are shown in Figure 10b. The maximum averaged signal intensity in the area-1 (${\bar{I}}_{Cu}$) was about 1670. On the other hand, that in-area-2 (${\bar{I}}_{exp}$) is about 300. We will use these values to determine the experimental attenuation distance $\Lambda_{exp}$ in the prepared concrete slab. In determining the attenuation distance $\Lambda_{exp}$, let the reflectance *R* of MM waves on the concrete surface be 0.22 and the reflectance of MM waves on the copper plate surface be *R*_Cu_. When the signal intensity transmitted from the Tx7 is *I*_0_, the received signal intensity ${\bar{I}}_{Cu}$ reflected directly from the copper plate can be expressed as follows:(11)$${\bar{I}}_{Cu}=R_{Cu}I_{0}.$$

It is assumed here that all returned signals are received. Using (11), the intensity ${\bar{I}}_{exp}$, which is the signal received after passing through the thin concrete slab twice, can be expressed as follows:(12)$${\bar{I}}_{exp}={\left(1-R\right)}^{2}\times {\left(exp\frac{-3.7}{\Lambda_{exp}}\right)}^{2}\times R_{Cu}I_{0}=0.78^{2}\times {\left(exp\frac{-3.7}{\Lambda_{exp}}\right)}^{2}\times {\bar{I}}_{Cu}$$

Using (11) and (12), $\Lambda_{exp}$ can be obtained as follows:(13)$$exp\frac{-3.7}{\Lambda_{exp}}=\frac{1}{0.78}\times \sqrt{\frac{{\bar{I}}_{exp}}{{\bar{I}}_{Cu}}}=\frac{1}{0.78}\times \sqrt{\frac{300}{1670}}=0.54$$
(14)$$\Lambda_{exp}=\frac{-3.7}{ln\left(0.54\right)}=6.0\;mm$$

The result obtained by this experiment is almost the same as the estimated value of Λ=5.4 mm, which indicates the difficulty in investigating deeper internal structures from the concrete surface.

However, the distance *L* of the target was limited to a maximum of about 3200 mm due to space limitations in the laboratory. In addition, only eight receive antennas were used to obtain better MM-wave images in the present study. Therefore, if we have a large laboratory space where we can observe at a distance of *L* = 10 m, and if we use all antenna elements of the array for measurements, we can expect a significant improvement in SNR. This would also make it possible to obtain transmission images of thicker concrete.

### 5.5. Evaluation of Transmission Performance Using Residential Wall Materials

As an important NDT technology in Japan, which is an earthquake-prone country, several observations have been made on the transmission performance of MM waves through composite plywood, refractory board, and ceramic tiles used for walls and other surfaces in residential houses.

Figure 11a shows the arrangement of the composite plywood and the copper plate placed at *L* = 3110 mm and *L* = 3225 mm, respectively. Figure 11b,c show the MM-wave images of the inside of the composite plywood at *L* = 3120 mm and the surface of the copper plate at *L* = 3225 mm, respectively. As can be seen from the MM-wave images, the composite plywood can transmit MM waves with almost no power loss when the thickness is about 12 mm. On the other hand, the faint streaks along the inserted arrows in Figure 11b become thicker and more distinct streaks in Figure 11c. Since there are no such streaks on both surfaces of the composite plywood, an inner sheet of the composite plywood may have some cracks or scratches. It is assumed that the scattering of MM waves caused by these cracks or scratches inside the composite plywood dramatically reduces the amount of MM-wave radiation to the copper plate directly behind the cracks or scratches.

Next, to confirm that MM waves are indeed effective in detecting damage as small as a few tens of µm, the detection performance was first evaluated using cracks that, unlike the concrete cracks in Figure 6, cannot be visually confirmed by humans at a distance. Figure 12a shows a refractory board of 5 mm width, which was completely cracked for use in this measurement. After splitting, the board was mounted so that the cracks were not visible, so it is difficult to see in this picture, but it can be seen that the crack runs down the left side. The width of the crack is only tens of µm in width, as shown in the inset. Figure 11b shows the MM-wave image observed on the surface of this refractory board. The ratio to the 300 mm wide board size shows that a crack structure of tens of µm width is observed as a crack image of about 8 mm width, which corresponds well to the actual crack position.

Now consider what happens when two boards are completely pasted together, since the range resolution $\delta_{L}$ of the FMCW radar is given by using the frequency bandwidth *B* = 4.5 GHz and the speed of light *c* as follows:(15)$$\delta_{L}\;≅\;\frac{c}{2B}=\frac{3.0\times 10^{8}}{2\times 4.5\;\times 10^{9}}\;\left[m\right]=33\;\left[mm\right].$$

This result means that even if a 2D slice image corresponding to a distance of *L* = 3080 mm is cut out from the range FFT analysis results, it will also be affected by objects in the range of 16.5 mm in front and behind it. Therefore, if this refractory board were transparent to MM waves, it would be affected not only by MM-wave scattering due to surface cracks but also by scattering due to internal cracks. This could therefore be observed as a larger signal change. Considering this, the penetration length of the concrete slab obtained this time was only about 6 mm, but it would be possible to detect cracks with higher sensitivity if even such a small penetration could be achieved.

Finally, the MM-wave transmission properties of ceramic materials such as ceramic tiles and a refractory board were investigated. Figure 13a,b show the photographs of 5 mm thick ceramic tiles bonded to the surface of a 5 mm thick refractory board with ceramic adhesives. The surface of the refractory board was engraved with various patterns for the transmission imaging. Figure 13c shows the MM-wave image at the interface between the ceramic tiles and the refractory board. It can be seen that the patterns on the refractory board are observed by transmission MM-imaging through the ceramic tiles. In addition, Figure 13d shows the MM-wave image of the copper plate placed at *L* = 3220 mm behind the ceramic tiles and refractory board. It can be seen that although the combined thickness of the ceramic tiles and refractory board is 10 mm, the reflective image of the copper plate is well observed, which also reflects the boundary lines between the tile boards and the pattern engraved on the refractory board.

Through these experiments, we were able to demonstrate that the high-sensitivity MM-wave imaging system we developed this time is very effective for NDT inspection of building materials such as residential houses.

## 6. Conclusions

In this study, we developed an MM-wave SIMO-SAR imaging system with a high-sensitivity MIMO radar module.

First, to evaluate the performance of the imaging system, the MM-wave reflection image of a copper plate was observed. As a result, a very high SNR value of 17 dB was obtained, reflecting the very low signal transmission loss between the array antenna elements and four cascaded chips in this high-sensitivity radar module.

Next, measurements were made on concrete materials to investigate the applicability to actual non-destructive testing of infrastructure facilities. It was found that by increasing the target distance *L* from 1000 mm to 3050 mm on an RC slab, MM-wave images that slightly reflected the interior steel frame were observed. However, it was not possible to observe the steel frame structure buried deeper from the concrete surface. Therefore, in order to obtain the attenuation distance in concrete, we prepared a thin concrete slab with a thickness of 3.7 mm and tried to observe the transmission images. As a result, we succeeded in experimentally obtaining the attenuation distance in the concrete of $\Lambda_{exp}$ = 6 mm.

In addition, transmission measurements were made on composite plywood, ceramic tiles, and a refractory board as examples of construction materials used in residential houses. MM-wave transmission was good for these materials, and even a copper plate placed behind these targets could be imaged transparently. In addition, we were able to detect areas of flaws that exist inside the material and cannot be seen visually.

Finally, in the inspection of infrastructure facilities, etc., a large area is almost always the target of inspection, and it is necessary to efficiently detect minute cracks and defects of tens of microns in size that occur near the surface of objects with such large areas. In general, inspection techniques using short-wavelength light such as visible light and X-rays tend to be superior to those using MM waves for detecting such minute defects. However, this is the case for target sizes ranging from a few millimeters to a few centimeters. When large areas of several meters need to be measured, such as infrastructure facilities, the low spatial resolution of MM-wave measurement comes into play. This is because line and point defects with a width and size of only a few tens of microns can be magnified to the size of several millimeters and detected when observed with MM waves. However, it is necessary to reliably observe a slight decrease in the received signal due to the scattering of the MM waves by these defects, etc., to achieve this. The MM-wave radar module of the present system was proven to have extremely high sensitivity characteristics that satisfy these conditions.

Non-destructive testing in various hazards is best performed from a distance to protect inspectors. In this sense, it is very important to develop an imaging system capable of making highly sensitive and accurate measurements from a distance. In this study, we successfully developed an MM-wave imaging system that satisfies these requirements.

## Figures and Tables

**Figure 1 sensors-24-04781-f001:**
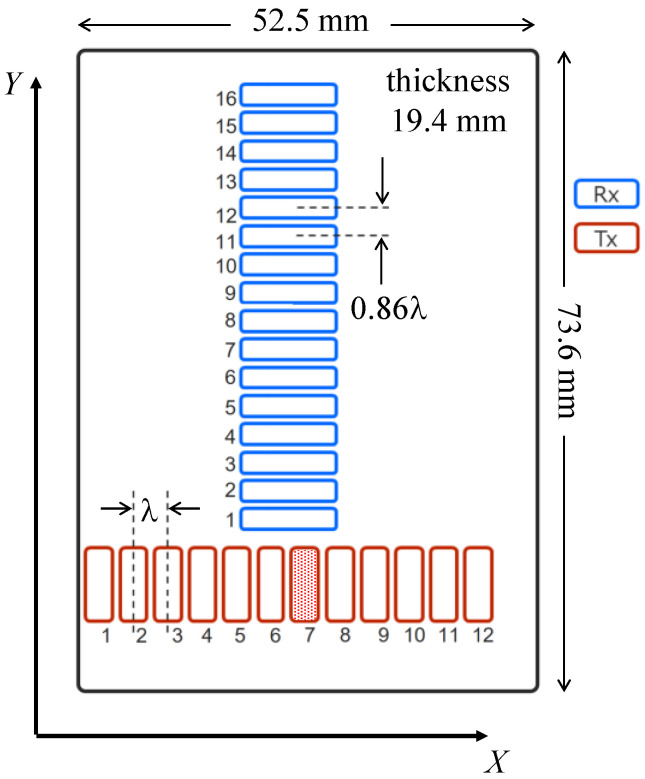
A 2-dimensional orthogonal arrangement of antenna elements of the array prepared for the present study. In the present experiments, only the seventh transmit antenna (Tx7) was used. Where *λ* = 3.80 mm.

**Figure 2 sensors-24-04781-f002:**
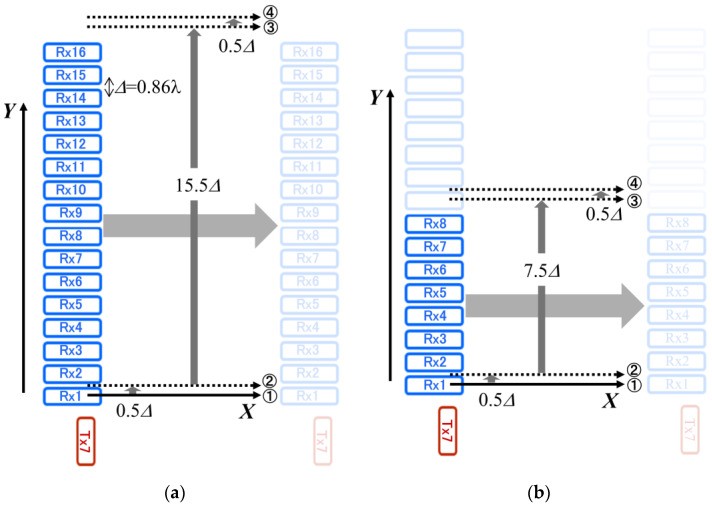
*X*-*Y* scanning method for antenna elements of the array. (**a**) Diagram of *X*-*Y* scanning using all Rx antennas from Rx1 to Rx16. (**b**) Diagram of *X*-*Y* scanning using eight Rx antennas from Rx1 to Rx8. Here, ∆ = 0.86 λ is the distance between adjacent receive antennas.

**Figure 3 sensors-24-04781-f003:**
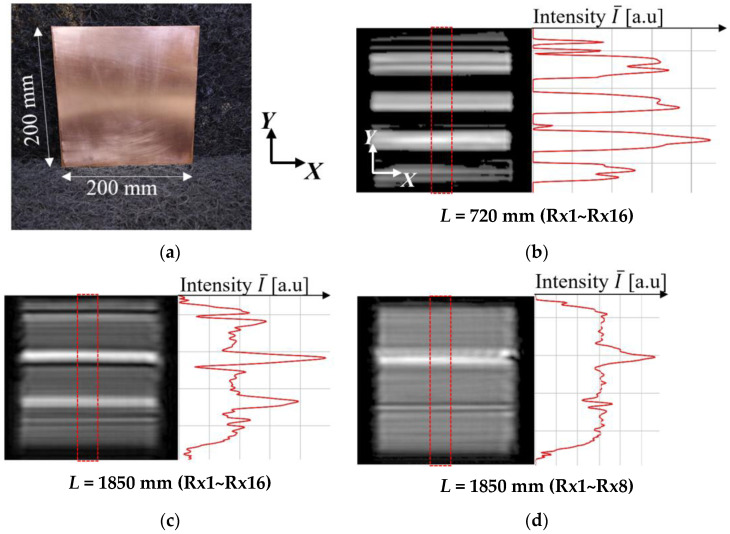
Influences of the distance *L* to the target and the number of receive antennas (Rx) used in the MM-wave image observations. (**a**) The 200 mm × 200 mm flat copper plate for the measurements. (**b**) Condition of *L* = 720 mm with all receive antennas from Rx1 to Rx16. The left and the right figures show the observed MM-wave image and the distribution along the *Y*-axis of the averaged signal intensity $\bar{I}$ within the red dotted rectangle inserted in the MM-wave image, respectively. (**c**) Result in the condition of *L* = 1850 mm with all receive antennas from Rx1 to Rx16. (**d**) Result in the condition of *L* = 1850 mm with 8 receive antennas from Rx1 to Rx8.

**Figure 4 sensors-24-04781-f004:**
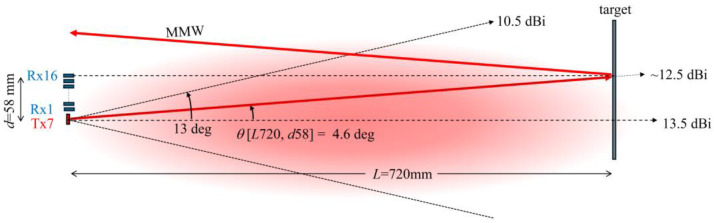
Estimation of MM-wave irradiation power on a target at a distance of *L* = 720 mm from the Tx antenna to the target.4.1. Optimization of MM-wave imaging measurements.

**Figure 5 sensors-24-04781-f005:**
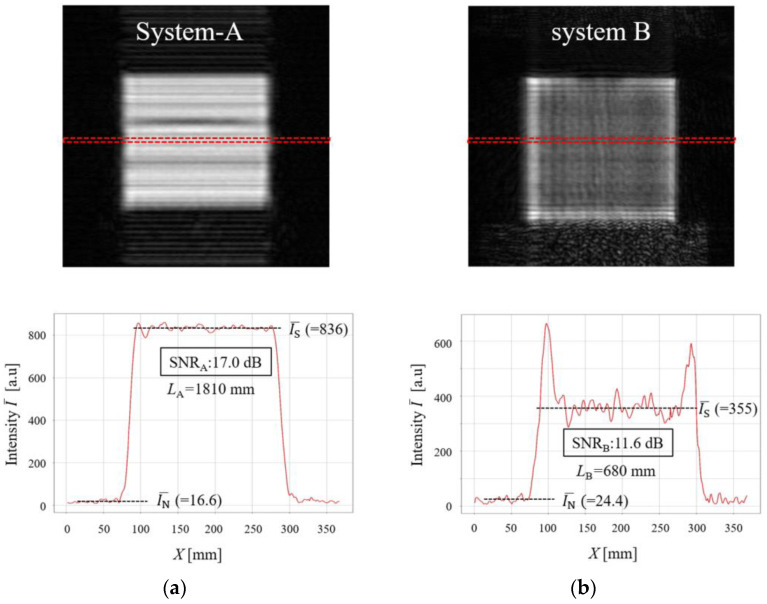
Comparison of SNR in the MM-wave images observed by (**a**) present SIMO-SAR imaging system (system-A) at *L*_A_ = 1810 mm and (**b**) previously constructed MIMO-SAR system using an IWR1443BOOST EVM (system-B) at *L*_B_ = 680 mm. The top figures show the MM-wave images, while the bottom figures show the profiles of the averaged signal intensity $\bar{I}$ along the *X*-axis direction in the area indicated by the red dashed rectangle inserted in the MM-wave image.

**Figure 6 sensors-24-04781-f006:**
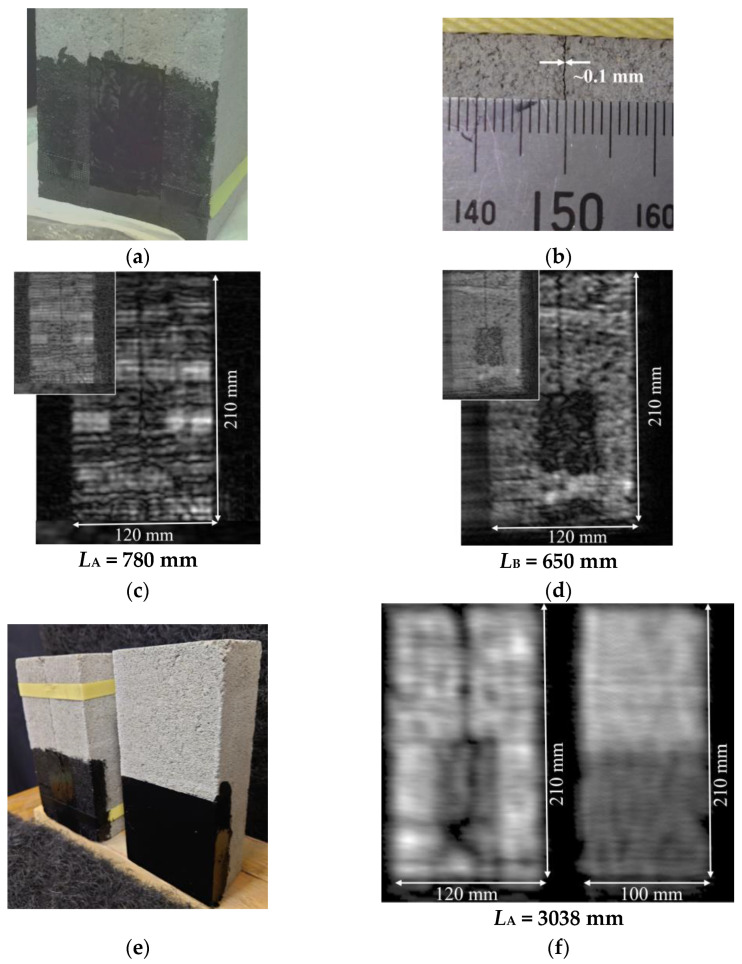
Comparison of MM-wave images of concrete blocks with GFR tape and black acrylic paint over the cracks. (**a**) Strongly bonded concrete blocks for observation. The contrast of the raw image is adjusted to show the position of the GFR tape under black acrylic paint. (**b**) Magnified view of the cracked area. It can be seen that the crack is about 0.1 mm wide. (**c**) MM-wave image observed at the target distance of *L*_A_ = 780 mm using system-A. (**d**) MM-wave image observed at the target distance of *L*_B_ = 650 mm using system-B. (**e**) Concrete blocks with (left) and without (right) cracks, thickly coated with acrylic paint over GFR tape. These were placed at a distance of *L*_A_ = 3038 mm from the radar module. (**f**) MM-wave images of concrete blocks with (left) or without (right) cracks.

**Figure 7 sensors-24-04781-f007:**
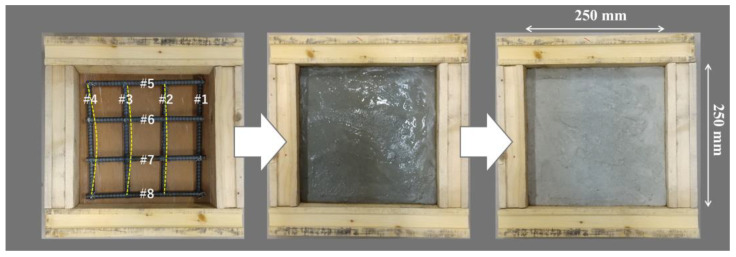
Reinforced concrete slab to investigate MM-wave transmission characteristics into concrete.

**Figure 8 sensors-24-04781-f008:**
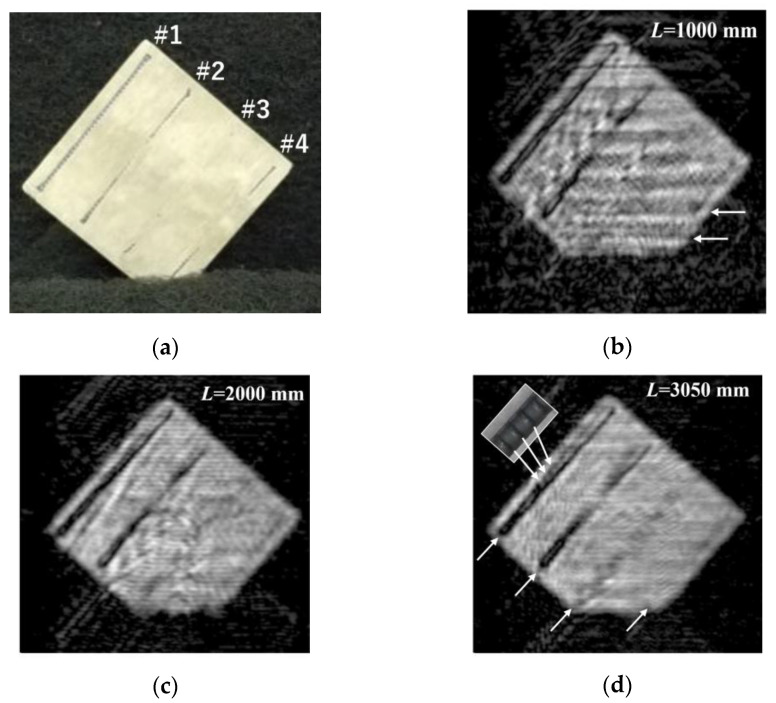
(**a**) Photograph of the prepared 33.4 mm thick concrete slab for MM-wave imaging. This surface was used for the measurement. (**b**) MM-wave image observed with the slab placed at a relatively short distance of *L* = 1000 mm. Below are MM-wave images observed at (**c**) *L* = 2000 mm and (**d**) *L* = 3050 mm.

**Figure 9 sensors-24-04781-f009:**
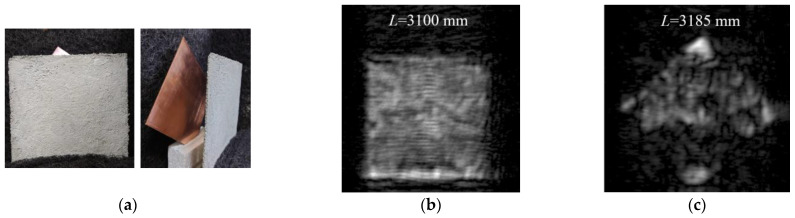
(**a**) Photographs of the thin concrete slab of 3.7 mm thickness placed at *L* = 3100 mm and the copper plate placed at *L* = 3185 mm behind it. (**b**) MM-wave image at position *L* = 3100 mm on the concrete surface and (**c**) MM-wave image at position *L* = 3185 mm on the copper plate surface.

**Figure 10 sensors-24-04781-f010:**
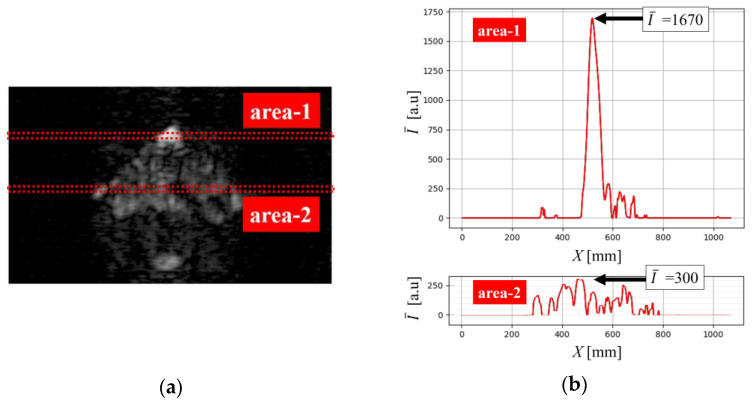
(**a**) area-1 and area-2 for comparing averaged signal intensity $\bar{I}$ in the MM-wave image. (**b**) Distribution of the averaged intensity $\bar{I}$ along the *X*-axis in area-1 and area-2.

**Figure 11 sensors-24-04781-f011:**
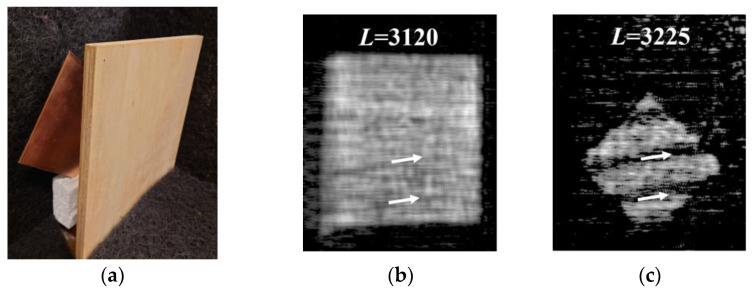
(**a**) A photograph of the composite plywood of 11.85 mm thickness placed at *L* = 3110 mm and the copper plate placed at *L* = 3225 mm behind it. (**b**) MM-wave image at position *L* = 3120 mm inside the composite plywood and (**c**) MM-wave image at position *L* = 3225 mm on the copper plate surface.

**Figure 12 sensors-24-04781-f012:**
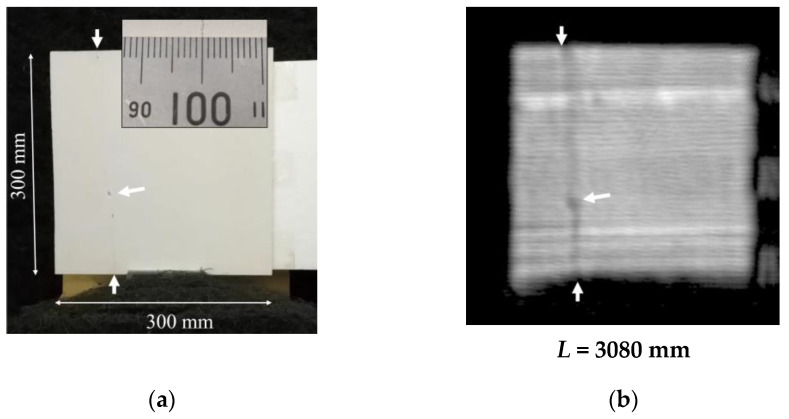
(**a**) A refractory board attached to a Styrofoam board with double-sided tape. It was pre-split and set at *L* = 3080 mm. The insert shows the cracks of tens of µm in width. The cracks cannot be seen from a distance. (**b**) MM-wave image of the refractory board at a distance of *L* = 3080 mm from the radar module. It can be seen that the crack streaks are imaged.

**Figure 13 sensors-24-04781-f013:**
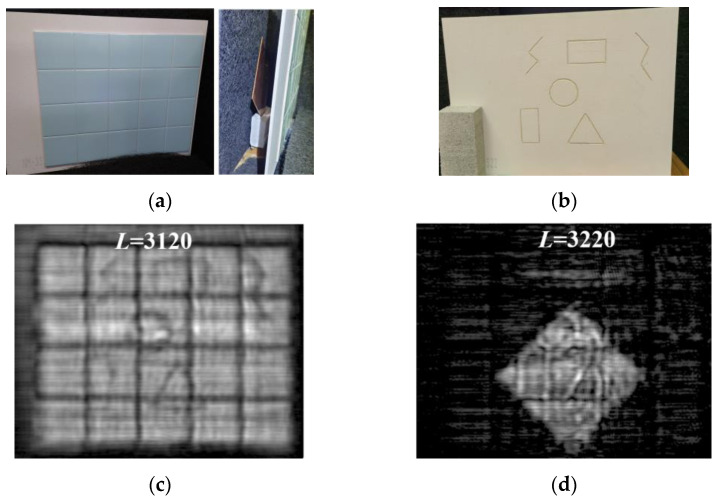
(**a**) Photograph of ceramic tiles and a refractory board attached with ceramic adhesives placed at *L* = 3115 mm and a copper plate placed at *L* = 3220 mm behind them for MM-wave transmission imaging. (**b**) Photograph of the pattern engraved on the surface of the refractory board bonded to the ceramic tile. (**c**) MM-wave image at the interface between ceramic tile and refractory board. The pattern engraved on the surface of the refractory board can be seen. (**d**) MM-wave image at *L* = 3220 mm on the copper plate surface.

**Table 1 sensors-24-04781-t001:** Element statistics of the MM-wave radar in the present study.

	Tx Antenna	Rx Antenna
Element interval	3.80 mm (1.00 λ)	3.25 mm (0.86 λ)
Number of elements	12 elements (in *X*-direction)	16 elements (in *Y*-direction)
Typical gain	*G*_Tx_ = 13.5 dB	*G*_Rx_ = 14.5 dB
Directivity: beam width	26 deg	22 deg
Directivity: side lobe	<25 dB	<20 dB
Frequency	76.5 GHz~81 GHz	
Antenna unit size (element + feed line)	52.5 mm × 73.6 mm × 19.4 mm	

## Data Availability

The dataset is available from the authors upon request.

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
