# Peer review of "Development of a High-Sensitivity Millimeter-Wave Radar Imaging System for Non-Destructive Testing"

_sensors, 2024, doi:10.3390/s24154781_

Round 1

Reviewer 1 Report

Comments and Suggestions for Authors

Biggest concern about the paper is if the topic is relevant and up-to-date. Paper provides 3 references regarding MM-wave NDT [13-15], where references [13] and [14] are 2006 and 2009 (i.e., 15 years old), and reference [15] has 2 common co-authors with the paper, which is not bad, but means that it is scientists from the same research group who are investigating MM-wave NDT. Is there anybody else interested in this topic? Why people stopped doing research in that area for 14 years? Is there any comparison study that concludes in which cases/scenarios MM-wave NDT might be better than other mentioned NDT methods?

Another (but seemingly related concern) is about the practicality of investigated method for concrete. According to obtained results, attenuation distance for concrete is about 6 mm. Is this a practically useful value for NDT scenarios? Experimental results also show that it is possible to detect cracks hidden by a paint and a fiber tape. Is that also a practical scenario? Which NDT scenarios are typically considered for other methods?

Also, is it possible to present some numerical metric for the performance of developed system? It is possible to see a crack on a Fig. 6 (b), however, that crack image is very vague, so one might argue that it is easy to confuse with something else. What might be better in such experiment is to have two samples, one with crack and one without, and be able to distinguish one from another in a blind test without prior knowledge that we are supposed to see a crack on the image. Also what might be better is to have some sort of comparison metric, which could be measured and used to tell that MM-wave NDT is better compared to some other NDT method.

Next is some minor concerns. Authors mention that they were able to detect streaks on composite plywood in their experiments (Fig. 11 (b) and especially (c)). However, there is two questions. 1) If these streaks correspond to the inner sheet, why are they even visible on the SAR image slice that is supposed to show reflective copper plate (Fig. 11 (c)) ? 2) Why not to check that these streaks are present indeed in the plywood, and not a result of some malfunction in the developed system.

Also, since authors use their setup to find out attenuation distance for concrete, why not to use the same setup to find attenuation distance for other two samples, plywood and ceramic? Experimental part for these samples have been done anyway.

Author Response

Dear Sir

Thank you very much for your valuable comments.

Our responses to your comments are as follows:

Comments1a: Biggest concern about the paper is if the topic is relevant and up-to-date. Paper provides 3 references regarding MM-wave NDT [13e15], where references [13] and [14] are 2006 and 2009 (i.e., 15 years old), and reference [15] has 2 common co-authors with the paper, which is not bad, but means that it is scientists from the same research group who are investigating MM-wave NDT. Is there anybody else interested in this topic? Why people stopped doing research in that area for 14 years?

Response 1a: Thank you for your comments.

You are absolutely right.

In response to your comments, we have excluded those that are less relevant to our non-destructive and non-contact testing, and added references to more relevant millimeter-wave-based studies. We have also added a detailed description of previous millimeter-wave inspection methods in the Introduction to provide more background for our research.

Lines 73-84

This time, we aim to develop a relatively compact, portable, non-contact, non-destructive, and remotely operable inspection system using a recently developed single-board millimeter-wave radar module.

The millimeter-wave module with transmit and receive antennas on a single board will not be commercialized and readily available to general researchers like us until around 2018, probably when it will be widely used in automotive safety driving systems. Until then, millimeter-wave signals were measured and analyzed using a combination of large, numerous, and expensive pieces of equipment, including signal generators, spectrum analyzers, power meters, horn antennas, and waveguides. This equipment was complex to install and operate and required a high level of expertise. With the advent of on-board millimeter-wave radar chips and millimeter-wave radar modules, these complex setups are no longer necessary, and measurements can be made easily and inexpensively.

Recently, some research has been conducted using SAR imaging systems with such millimeter-wave radar modules, but commercially available products have weak antenna gain and cannot be used for nondestructive inspection. Therefore, in cooperation with Taiyo Yuden Co., LTD, we developed a radar module with extremely low signal transmission loss, high directivity, and high gain. As a result, there is hardly any other non-destructive inspection that uses such a board.

Comments1b: Is there any comparison study that concludes in which cases/scenarios MM-wave NDT might be better than other mentioned NDT methods?

Response 1b:  Each NDT method has its advantages and disadvantages. Therefore, we do not believe it is useful to make a general comparison of them.

One of the advantages of the millimeter-wave inspection method is that the effect of a tiny defect as small as 0.1 mm, as in this case, appears over a measurement wavelength λc range of several mm. In nondestructive inspection of a sample with a relatively large area of several meters, the effect of submicron damage appears as an anomaly of a few millimeters on the observed image, making it easy to detect. This is because in inspections using visible light or X-rays, which have short wavelengths and enable high-resolution observation, it is considered extremely difficult to find damage of only 0.1 mm (~1/10000) in the observation image of several meters in size. After all, the effect appears only at the actual 0.1 mm level.

In order to detect such effects with high sensitivity in millimeter-wave measurements, the radar module with very high directivity and high antenna gain that we have developed is a very important factor. These are described in detail in sections 5.1 and 5.5(The text is colored yellow in the sections).

Comments 2: Another (but seemingly related concern) is about the practicality of investigated method for concrete. According to obtained results, attenuation distance for concrete is about 6 mm. Is this a practically useful value for NDT scenarios? Experimental results also show that it is possible to detect cracks hidden by a paint and a fiber tape. Is that also a practical scenario? Which NDT scenarios are typically considered for other methods? 

Response 2:Thank you for your comment.

First of all, I think the only way to try to observe the internal conditions of reinforced concrete is to use radiation such as X-rays or neutron beams. In this sense, we don't think that a 6 mm penetration length is useful to observe the conditions inside the concrete. I think we can only inspect the condition near the surface. However, we are planning to try to measure with all transmit and receive antennas in a larger laboratory space (L=8-10m), in which case the gain of the number of arrays (Rx: 16ch, Tx: 12ch) alone will be about 22dB, which is quite high when combined with the gain of the transmit and receive antennas. Therefore, if the gain of the number of arrays (Rx: 16ch, Tx: 12ch) is considered together with the gain of the transmit and receive antennas, a much higher sensitivity can be expected and observation with a higher SNR should be possible. The attenuation length Λ is the distance over which the intensity of the incident electromagnetic wave becomes e-1, so a return trip of Λ will attenuate the incident wave by about 13%. This is just my expectation, but I believe that even if the attenuation is reduced by a factor of 100 or even less, it will be possible to detect the received signal if the SNR improves by that amount. In addition to this inspection of the internal structure, what is more important in concrete inspection is the early detection of small cracks near the surface and their proper repair. In this sense, if near-surface cracks can be detected, it will be an effective inspection tool.

Furthermore, if the range resolution approach discussed in Section 5.5 for the results of Figure 12 is applied, it is possible that cracks and internal damage near the surface will be easier to find if they penetrate into the interior and are returned as reflections.

On the other hand, concrete has been reinforced with glass fiber to strengthen the concrete, and the surface is often coated with a protective coating to protect the interior. The test was conducted by coating the surface with glass fiber-reinforced tape and acrylic paint for corrosion protection.

Comments 3: Also, is it possible to present some numerical metric for the performance of developed system? It is possible to see a crack on a Fig. 6 (b), however, that crack image is very vague, so one might argue that it is easy to confuse with something else. What might be better in such experiment is to have two samples, one with crack and one without, and be able to distinguish one from another in a blind test without prior knowledge that we are supposed to see a crack on the image. Also what might be better is to have some sort of comparison metric, which could be measured and used to tell that MM-wave NDT is better compared to some other NDT method.

Response 3: Thank you for your comment.

I agree that the image may be difficult to understand. I measured the image again at about L=3000mm, which is the condition where the highest resolution image can be obtained in this experimental space.[Figure 6(e) and (f)]

With the previous L=780mm, as I commented in the paper, the difference in distance from the transmitting antenna affects the received signal strength of each receiving antenna, resulting in a stripe structure in the horizontal direction. This is not the case in the replaced image, so you can see the cracks. In addition, we also observed a concrete block without cracks that had been coated with fiber-reinforced tape and protective acrylic paint, which we have also included here. 

I also took a magnified photo of the cracked area to give you an idea of the width of the crack, just in case.

As discussed in the paper [Section 5.1], distance decreases resolution in SAR imaging. However, this is also the case because crack structures with small widths can appear as cracks with large widths, and we believe that such observations at greater distances are very important in terms of saving the lives of the observers.

Comments 4: Next is some minor concerns. Authors mention that they were able to detect streaks on composite plywood in their experiments (Fig. 11 (b) and especially (c)). However, there is two questions. 1) If these streaks correspond to the inner sheet, why are they even visible on the SAR image slice that is supposed to show a reflective copper plate (Fig. 11 (c)) ? 2) Why not to check that these streaks are present indeed in the plywood, and not a result of some malfunction in the developed system.

Response 4: Thank you for your comment.

We determined that there is some kind of line defect inside the plywood, because the locations of the streaks along the arrows inside the plywood and the streaks visible on the copper plate are the same, and the image on the copper plate is much darker with much lower signal intensity.

1) why are they even visible on the SAR image slice that is supposed to show a reflective copper plate?

The answer to this question is as follows.

If there is some kind of defect inside the plywood, millimeter waves will be scattered at that location. This scattering is greater than in other normal areas, and as a result, the intensity of the millimeter waves traveling through that area to the copper plate is reduced. The millimeter waves are then scattered again at approximately the same location as they are reflected from the copper plate, resulting in a weaker received signal strength. This is why streaks appear in the image on the copper plate.

When the measurement distance is L=3200mm, there is an angle of about 1.25 degrees in the elevation direction between the incident wave radiated on the copper plate and the received reflected wave from the copper plate surface, which causes wider streaks on the copper plate than on the plywood.

2) Why not to check that these streaks are present indeed in the plywood?

As you pointed out, we also tried to check by actually peeling off the plywood but stopped when we realized that it was impossible to peel off the plywood in one cut.

Furthermore, we have done many experiments with similar plywood, but we have never observed such diagonal streaks.

Comments 5: Also, since authors use their setup to find out attenuation distance for concrete, why not to use the same setup to find attenuation distance for other two samples, plywood and ceramic? Experimental part for these samples have been done anyway.

Response 5: Thank you for your valuable comment.

Ceramic tiles and refractory board are standard materials used in actual houses, and we did not observe attenuation lengths for these because they were found to be permeable.

As for wood, we are interested in studying it because we would like to observe the structure of nests formed inside old trees, etc., but we cannot obtain this information by calculation because we do not know the refractive index, as we first assumed for concrete. Another important point is that experiments have shown that the transmission characteristics vary considerably depending on the moisture content, so at this stage we are studying different conditions.

Reviewer 2 Report

Comments and Suggestions for Authors

Authors present a design of a MMW radar imaging system for non-destructive testing. The manuscript is well written and easy to follow. This reviewer's comments are as follows. Please address the following issues in the revised manuscript.

1. Lines 112-114:

How do you combine the first pass data with the next one with 0.5 delta shift in the Y-axis direction? Please provide some concept or theory for this method.

2. Figure 11:

To obtain Figures 11b and 11c, do you apply some kind of range gating so that you can see the image of the layer of your choice?

3. Figure 12c:

The resolution in x and the one in y direction seem to be roughly the same.

1) Could you provide some kind of estimate the radar resolution in the x direction and in the y direction?

2) Is the resolution in the x direction determined by the principle of Doppler sharpening?

3) What is the mechanism behind the good resolution in the y direction?

Author Response

Dear Sir

Thank you very much for your very positive comments.

The followings are our responses.

Comments 1: Lines 112-114. How do you combine the first pass data with the next one with 0.5 delta shift in the Y-axis direction? Please provide some concept or theory for this method.

Response 1: Thank you for your comment.
We use Matlab. to process the data. After the measurement, We sort the obtained data in the Y-direction for SAR analysis.

Comments 2: Figure 11. To obtain Figures 11b and 11c, do you apply some kind of range gating so that you can see the image of the layer of your choice?

Response 2: Thank you for your comment.

Yes, you are right. We reported on our image processing technique using Range FFT data last year at IEEE Access, so you can refer to that as well [1]. Among the data processing in that report, we are using 2D slice images resolved by distance in this case. From the Range FFT data obtained by analysis, all signal intensities corresponding to the distance L to be extracted are taken at each (X,Y) coordinate and mapped in two dimensions.

[1]  Wang, Y.; Su, J.; Fukuda, T.; Tonouchi, M.; Murakami, H. Precise 2D and 3D Fluoroscopic Imaging by Using an FMCW Millimeter-Wave Radar. IEEE Access. 2023, 11, 84027-84034. 

Comments 3: Figure 12c. The resolution in x and the one in y direction seem to be roughly the same.

1) Could you provide some kind of estimate the radar resolution in the x direction and in the y direction?

2) Is the resolution in the x direction determined by the principle of Doppler sharpening?

3) What is the mechanism behind the good resolution in the y direction?

Response 3: Thank you for your comments. The resolution of imaging by SAR technology is mostly determined by the scan width and the distance to the object to be measured. The sampling interval al The same principle applies not only to data in the x-direction but also to data in the y-direction, which are sorted and analyzed after all the data have been obtained.

However, in the case of our device, δy cannot be smaller than Δ/2, considering the step width.

The resolution of our system is additionally described in the Lines 309-366.

[2] Juan M. Lopez-Sanchez et al, "3-D Radar Imaging Using Range Migration Techniques," IEEE Access, (2000) 48, pp. 728-737.

In the attached PDF file, we have also colored the additions in yellow to reflect other reviewers' comments.

Reviewer 3 Report

Comments and Suggestions for Authors

The author has developed a single-input and multiple-output (SIMO) synthetic aperture radar (SAR) imaging system using a high-sensitivity millimeter-wave (MM-wave) array antenna module, which is interesting. However, there are still some issues that need to be addressed before publication.

1.     The author should add a comparison with the key indicators of relevant research in the abstract to highlight the advantages of the proposed scheme, which will also be more convincing.

2.     The author should analyze whether this plan can contribute to key indicators and low angle resolution

Author Response

Dir Sir

Thank you for your valuable comments.

The followings are our responses to your comments.

Comments 1: The author should add a comparison with the key indicators of relevant research in the abstract to highlight the advantages of the proposed scheme, which will also be more convincing.

Comments 2: The author should analyze whether this plan can contribute to key indicators and low angle resolution

Response 1 & 2:

Thank you for your comment.

Each NDT method has its advantages and disadvantages. Therefore, we do not believe it is useful to make a general comparison of them, but we believe it is important to develop a highly sensitive inspection method that can detect sub-millimeter cracks and internal damage near the surface by non-destructive and non-contact inspection even at a distance. This is because when inspecting buildings that may collapse after an earthquake or other disaster, it is necessary to protect the lives of the technicians performing the inspection, and it is desirable to be able to inspect, to some extent, slight cracks and internal surface damage that cannot be seen from a distance.

We are currently conducting research to contribute to the development of such inspection equipment. Therefore, we do not directly compare the specifications with those of other inspection systems, but we have described our main indexes and the degree of achievement against them in as much detail as possible in the Abstract, Introduction of this paper, and each Section for the results of each experiment.  

As a result of this experiment, the low resolution of millimeter waves with long wavelengths (compared to light and radiation), which we initially thought was a disadvantage, was reversed and the results showed the effectiveness of millimeter waves compared to other measurement methods, and we can understand how important it is to develop a millimeter wave imaging system with higher sensitivity than spatial resolution. We believe that the effectiveness of the high directional and high sensitivity millimeter-wave imaging system we have developed this time has been fully confirmed.

We are very sorry we don’t understand “contribute to low angle resolution” in comments 2.

If you are referring to angular resolution in millimeter-wave imaging, it can be sufficiently improved in the future.

The angular resolution of MIMO radar is inversely proportional to the number of antennas, so using 16 Rx antennas instead of the 8 currently used will improve the resolution by a factor of 2, and using all the Tx7 transmit antennas, which are currently the only ones used, will improve it by a factor of 12.

Reviewer 4 Report

Comments and Suggestions for Authors

In this paper, a single input multiple output (SIMO) synthetic aperture radar (SAR) imaging system is developed, which uses a high sensitivity millimeter wave (MM wave) array antenna module. The module has high directivity and high signal-to-noise ratio, so it is a good realization of non-destructive testing of infrastructure structure. In general, this is a more detailed and interesting paper. Before publication, I think there are the following points to be improved.

1. Multiple articles were cited at one time, but their contributions or relevance to their own research were not specified (e.g. [3-9]). It is recommended to improve the relevant description or reduce unnecessary citations.

2. At the end of the introduction, it is best to add the introduction of chapter structure and main content.

3. The template of Table 1 is best to use the three-line table format.

4. Please put the sequence number (a), (b) and so on of the subgraph in Figure 3 directly below the corresponding image.

5. Note the line spacing between 109 lines to 110 lines, so that the full text remains unified, the following is the same.

Author Response

Comments1: Multiple articles were cited at one time, but their contributions or relevance to their own research were not specified (e.g. [3-9]). It is recommended to improve the relevant description or reduce unnecessary citations.

Response 1: Thank you for your comment. We agree with this comment. 

We have excluded those that are less relevant to our non-destructive and non-contact testing, and added references to more relevant millimeter-wave-based studies. We have also added a detailed description of previous millimeter-wave inspection methods in the Introduction to provide more background for our research.

Lins 73-84

Comments 2: At the end of the introduction, it is best to add the introduction of chapter structure and main content. 

Response 2: Thank you for your comment. We agree with this comment. 

We have added the main contents and results of this study at the end of the introduction.

Lines 82-101

Comments 3: The template of Table 1 is best to use the three-line table format.

Response 3: Thank you for your comment. We agree with this comment. 

We have reworked Table 1 into a three-column format.

Line 119 

Comments 4: Please put the sequence number (a), (b) and so on of the subgraph in Figure 3 directly below the corresponding image.

Response 4: Thank you for your comment. We agree with this comment. 

We have moved the sequence number for each image in Figure 3 directly below the image.

Figure 3

Comments 5: Note the line spacing between 109 lines to 110 lines, so that the full text remains unified, the following is the same.

Response 5: Thank you for your comment. We agree with this comment.

I can't be sure, but I think I checked and fixed the line spacing and the position of the figures.

In the attached PDF file, we have also colored the additions in yellow to reflect other reviewers' comments.

Round 2

Reviewer 1 Report

Comments and Suggestions for Authors

No major concerns left, but I would suggest revising text (especially, new text) one more time; in particular, abstract has "As a result, attenuation distance = 6 mm was obtained" repeated twice in a row.

Author Response

Comment: No major concerns left, but I would suggest revising text (especially, new text) one more time; in particular, abstract has "As a result, attenuation distance = 6 mm was obtained" repeated twice in a row.

Response: Thank you veru much. We have corrected the points you pointed out in this time.

Dear Sir

We would like to thank you for taking time out of your very busy schedule to peer review our manuscript.
Your valuable comments have helped us to improve the content of our paper.

We hereby express our sincere gratitude.

Hironaru Murakami
